# Design, Development, and Optimisation of Smart Linker Chemistry for Targeted Colonic Delivery—In Vitro Evaluation

**DOI:** 10.3390/pharmaceutics15010303

**Published:** 2023-01-16

**Authors:** Heba S. Abd-Ellah, Ramesh Mudududdla, Glen P. Carter, Jonathan B. Baell

**Affiliations:** 1Department of Medicinal Chemistry, Monash Institute of Pharmaceutical Sciences, Parkville, VI 3052, Australia; 2Department of Medicinal Chemistry, Faculty of Pharmacy, Minia University, Minia 61519, Egypt; 3Department of Microbiology and Immunology, The University of Melbourne at the Peter Doherty Institute for Infection and Immunity, Parkville, VI 3001, Australia; 4School of Pharmaceutical Sciences, Nanjing Tech University, No. 30 South Puzhu Road, Nanjing 211816, China; 5Institute of Drug Discovery Technology (IDDT), Ningbo University, 818 Fenghua Road, Ningbo 315211, China

**Keywords:** colonic delivery system, pH-responsive systems, cyclisation-activated prodrugs, mesalamine

## Abstract

Drug targeting is necessary to deliver drugs to a specific site of action at a rate dictated by therapeutic requirements. The pharmacological action of a drug can thereby be optimised while minimising adverse effects. Numerous colonic drug delivery systems have been developed to avoid such undesirable side effects; however, these systems lack site specificity, leaving room for further improvement. The objective of the present study was to explore the potential of amino-alkoxycarbonyloxymethyl (amino-AOCOM) ether prodrugs as a general approach for future colonic delivery. To circumvent inter- and intra-subject variabilities in enzyme activities, these prodrugs do not rely on enzymes but rather are activated via a pH-triggered intramolecular cyclisation–elimination reaction. As proof of concept, model compounds were synthesised and evaluated under various pH conditions, simulating various regions of the gastrointestinal tract (GIT). Probe **15** demonstrated excellent stability under simulated stomach- and duodenum-like conditions and protected 60% of the payload in a small intestine-like environment. Moreover, **15** displayed sustained release at colonic pH, delivering >90% of the payload over 38 h. Mesalamine (Msl) prodrugs **21** and **22** were also synthesised and showed better stability than probe **15** in the simulated upper GIT but relatively slower release at colonic pH (61–68% of Msl over 48 h). For both prodrugs, the extent of release was comparable to that of the commercial product Asacol. This study provides initial proof of concept regarding the use of a cyclisation-activated prodrug for colon delivery and suggests that release characteristics still vary on a case-by-case basis.

## 1. Introduction

Targeting drugs to the colon is highly desirable for treating a variety of bowel diseases, including Crohn’s disease, ulcerative colitis, amoebiasis, and colon cancer [1,2,3,4,5]. It can also be used for systemic delivery of labile molecules such as therapeutic peptides and proteins [6,7]. Colonic delivery can be achieved via the oral or rectal routes [8], although the oral route is usually preferred due to greater patient acceptance, relative manufacturing simplicity, and superior safety and efficacy [9]. The development of a colon-specific oral formulation is challenging as it must pass through the upper alimentary canal intact, avoiding any potential metabolism or drug release in the gastrointestinal tract (GIT) prior to releasing the drug in the proximal colon. Various colonic delivery systems have been developed to overcome these challenges, including the use of time-dependent formulation, pH-sensitive and bacterial degrading coating materials, hydrogels, biodegradable polymers, and prodrugs [10,11,12,13,14]. Each of these approaches comes with its own particular advantages and limitations [2].

The prodrug approach to colon targeting is plausible because of the clinical success of azo-prodrugs such as ipsalazide, olsalazine, and balsalazide in treating colonic conditions [15,16,17]. These prodrugs are activated by azoreductases produced by the colonic microflora. Nevertheless, the use of the azoreductase approach has been restricted to drugs bearing a primary aromatic amino group such as mesalamine (Msl). Previous studies have shown that double prodrug designs such as **1 (a–b)** can extend the possibilities for azoreductase-triggered drug release to alcohol-containing drugs (prednisolone) and benzenesulfonamide COX-2 inhibitors (celecoxib) [18,19]. With a propionic acid linker, **1 (a–b)** first undergoes reduction by colonic azoreductases, releasing latent prodrugs that spontaneously cyclise, liberating the parent drugs, as shown in Figure 1. Prodrug **1a** was as efficacious as prednisolone in a mouse model of ulcerative colitis with fewer systemic adverse effects [18]. Moreover, transepithelial diffusion of prodrug **1b** was effectively suppressed in the Caco-2 model compared to celecoxib, also indicating suitable characteristics for colonic delivery [19]. Similarly, a nitroreductase-triggered cyclisation approach was applied to prednisolone and celecoxib to produce prodrugs that were capable of releasing the parent drugs in an in vitro model [20].

Given that cyclisation-activated prodrug designs have been identified as a superior strategy to avoid inter- and intra-subject variability in enzymatic activities [21], the purpose of this paper is to further extend the use of an intramolecular cyclisation approach as a vector to trigger colonic drug release. Previous work has demonstrated that functionalising the nonsteroidal anti-inflammatory drug piroxicam with an ethoxycarbonyloxymethyl moiety results in a prodrug (ampiroxicam; Figure 1) that is stable at pH 1 (t_½_ ≥ 50 h) but hydrolyses slowly (t_½_ ≥ 8 h) at pH 6.0–8.5 [22]. Incorporation of an amino group into the alkoxycarbonyloxymethyl (AOCOM) substructure would permit release of the parent drug over a different pH range than for AOCOM. For example, this approach has recently been applied to develop an amino-AOCOM-containing prodrug **2**, which is stable under stomach-like conditions but cyclises to release the drug upon reaching the small intestine (Figure 1) [21]. In addition, **2** displayed improved aqueous solubility and enhanced oral bioavailability compared with the parent drug [21]. Herein, we present the amino AOCOM ether moiety, for the first time, as a pH-activated prodrug handle for colonic delivery. Since we required a prodrug that would be stable in the upper GIT but break down optimally in the colon, we focused on controlling the cyclisation rate by varying the linker, or spacer, carbon length of the amino-AOCOM moiety. Part one of our study was dedicated to the use of two fluorophores—4-carboxy-2-methyl Tokyo Green (CTG) and dicyanomethylene-4*H*-pyran dye (DCM-OH)—as model compounds to provide initial proof of concept and to study the effect of changing the chemistry of the payload on the release rate. Part two focused on applying the same concept to a representative drug used for colonic diseases (Msl) to maximise its efficacy in treating these conditions.

## 2. Materials and Methods

### 2.1. Synthetic Procedures and Analytical Data

All solvents and reagents were purchased from commercial sources, stored appropriately, and used as supplied. Reactions were monitored by TLC using silica gel on 60 F_254_-coated aluminium plates (0.25 mm thickness). TLC plates were visualised under UV light at 254 and 366 nm and with KMnO_4_ staining when required. NMR spectra were recorded using a Bruker spectrometer (^1^H, 400.13 MHz; ^13^C, 100.62 MHz). Chemical shifts were measured in parts per million (ppm) using tetramethylsilane as an internal reference in either CDCl_3_ (7.26 ppm for ^1^H and 77.16 ppm for ^13^C), MeOD (3.31 ppm for ^1^H and 49.00 ppm for ^13^C), or DMSO-d_6_ (2.50 ppm for ^1^H and 39.52 ppm for ^13^C) as solvent. Where possible, proton and carbon signals were assigned with the aid of 2D NMR experiments (HSQC and HMBC). The (*) symbol is used to indicate ^13^C signals that were only identified by 2D NMR experiments. The ESI–MS analyses were performed using an Agilent UHPLC/MS (1260/6120) system, whereas HRMS analyses used an Agilent 6224 TOF–MS instrument. Detailed synthetic procedures for the final probes **13**–**17** and Msl prodrugs **21** and **22** are provided in Appendix B.

### 2.2. In Vitro Release Assays

Test solutions for probes **13**–**17** were prepared at 25 µM in a 1:1 mixture of ethanol and the respective buffer. Samples were incubated at 37 °C, then analysed by HPLC using an Agilent 1260 Infinity system, with conditions as follows: Agilent ZORBAX Eclipse Plus C18 Rapid Resolution column, 3.5 μm (4.6 × 100 mm), 95 Å pore size; column temperature, 35 °C; injection volume, 2 μL; flow rate, 1.0 mL min^−1^; gradient elution from 5–100% solvent B (0.1% TFA in MeCN) in solvent A (0.1% TFA in ultrapure water) over 9 min, then hold for 1 min at 100% B; UV detection at two wavelengths (254 and 214 nm). The total run time was 10 min, and the retention times of the model compounds CTG and DCM-OH were 4.28 and 6.16 min, respectively. The retention times of probes **13**–**17** were 4.41, 4.53, 4.58, 5.97, and 6.04 min, respectively. Calibration curves of free fluorophore were constructed at pH values of 1.2, 5.0, 6.5, and 7.5 over the concentration range of 10–100 µM. A concentration of 25 μM was selected to run in the assay as it gave a good instrument response at the selected wavelength. The data at each pH value was fitted by linear regression, yielding equations of the form: y = ax ± b, where y is the peak area and x is the corresponding concentration in µM. Furthermore, the correlation coefficient (R^2^) was ≥0.999 in each case, confirming the linearity of the method used under the specified conditions. Similarly, Msl prodrugs **21** and **22** were dissolved in 1:1 water/buffer at a concentration of 250 µM, incubated as above, and analysed by the same HPLC method but using different detection wavelengths (310 and 254 nm). During assay development, the wavelength of 310 nm was found to be the most suitable for these compounds, as it is far away from the UV cut-off wavelength of the mobile phase, and the reported value is >260 nm [23], as described in detail in the Appendix A. The retention times of Msl and prodrugs **21** and **22** were 1.19, 1.38, and 1.39 min, respectively. Calibration curves of free Msl were constructed at the same pH values as above, over the concentration range of 50–1000 µM. The data were fit by linear regression as before, and a similar goodness of fit (R^2^ ≥ 0.999) was achieved in each case.

## 3. Results and Discussion

### 3.1. Proposed Release Mechanism

The proposed mechanism of drug release from the newly designed prodrugs, based on fundamental chemical principles and supported by related literature [21], is shown in Figure 1. The design mainly reflects the pH differences along the GIT. The pH in the stomach increases progressively from 1–2 during fasting to reach up to 5 after eating, and then the pH in the duodenum and the small intestine are 5 and 6.5–7.5, respectively. After that, the pH in the colon drops slightly, to 6.4–7.0 [2,24]. We believed that the prodrug would be stable in the stomach, where the amino functional group should be protonated to give a stable ionised system (that is, it will be non-nucleophilic). As the compound progresses through the GIT, there will be a gradual increase in pH. Once it reaches pH > 5, ammonium group deprotonation will initiate, causing subsequent intramolecular cyclisation (Figure 1) to release the drug along with side-products **3** (cyclic carbamate) and **4** (formaldehyde) [21], both of which are non-toxic at low concentrations [25]. The ease of cyclisation and the size of the lactam ring formed depend on the length of the methylene spacer. The amino-AOCOM linkers with two- and three-methylene spacers reportedly cyclise efficiently to give five- and six-membered rings, respectively [21], although cyclisation to a seven-membered ring has apparently not yet been explored. Furthermore, with the pH difference between the small intestine and the colon not being very pronounced, there is the potential for prior release of the drug in the small intestine. However, this may not be a problem if the release is slow enough to prevent too much payload loss in the small intestine, given the transit time in the small intestine (2 h) [26] is much shorter than that in the colon (30–40 h) [27,28]. Therefore, we examined amino-AOCOM handles with different methylene spacer lengths—as in **13**, **14**, and **15**—in an attempt to find a suitable pH-sensitive linker that would only cyclise in the colon to release the drug.

### 3.2. Synthesis

To synthesise probes **13–17**, the CTG-*tert*-butyl ester and DCM-OH fluorophore were first constructed as described previously [29,30,31], and then the linker was introduced as shown in Figure 2. Briefly, Boc-protected amino alcohols **8a–c** were prepared from pent-4-yn-1-ol and converted into AOCOM iodides **10a–c**. The reaction of *tert*-butyl CTG and DCM-OH with **10a–c** in the presence of Cs_2_CO_3_ afforded fluorophore conjugates **11a–c** and **12a–b**, which were deprotected using 32% HCl to yield probes **13–17**. The identities of probes **13–17** were confirmed by LC/MS and ^1^H NMR analysis, which showed the disappearance of *tert*-butyl signals and the appearance of two proton signals at 9.19, 8.96, 8.55, 8.98, and 8.86 ppm, respectively, corresponding to the ammonium salt of the linker, as shown in the Appendix A. The synthetic route to the Msl prodrugs commenced with the two-step protection of reactive groups (NH_2_ and COOH), as illustrated in Figure 3. First, Msl was reacted with Boc_2_O under basic conditions to give compound **18** in high yield, which was esterified with *tert*-butanol in the presence of CDI and DBU to afford compound **19**. Reaction of AOCOM iodides **10b–c** with **19** in the presence of Cs_2_CO_3_ gave compounds **20a–b**, which were deprotected under acidic conditions to give the final Msl prodrugs **21** and **22**. Proof of structure of **21–22** was gained from LC/MS and ^1^H-NMR analysis that showed one singlet at 5.87 and 5.86 ppm ascribed to the methylenedioxy group, respectively, and integrating three aromatic protons for Msl. Furthermore, the presence of a doublet of doublet signals of eight protons ascribed to the aromatic protons of two molecules of tosylic acid confirms that **21–22** were formed in the disalt form, as shown in Appendix A.

### 3.3. In Vitro Release of Probe Compounds

Release assays for probes **13–17** were performed in vitro using pH and temperature conditions chosen to simulate the GIT, with analysis by HPLC. We used KCl/HCl buffer (pH 1.2) and acetate buffer (pH 5.0) to simulate the pH of the stomach and the duodenum, respectively, while PBS (pH 7.5) and phosphate buffer (pH 6.5) were used to mimic the pH of the lower small intestine and the colon. The compounds were dissolved in 1:1 ethanol/buffer and incubated at 37 °C. Aliquots were taken at different time points and analysed by HPLC and LC/MS, as described in Section 2.2. Probes **13**,**16** (with a two-methylene spacer) and **14, 17** (with a three-methylene spacer) were examined first. The stability of the prodrug at low pH is a critical aspect of our design since there should be no (or minimal) drug release in the stomach during the normal gastric transit time of 3–5 h [26]. Thus, the aforementioned compounds were incubated at pH 1.2 for 5 h to test their stability under conditions simulating the stomach. All of the probes showed good stability, although compounds **14, 17** (three-methylene spacer) were slightly more stable. The measured fluorophore release was 6%, 2%, 3%, and 0.5%, respectively, for probes **13**, **14**, **16**, and **17**, confirming their expected stability at low pH (Figure 2A). Then, the stability of the probes was tested in simulated duodenum conditions for 30–40 min, reflecting the reported transit time in the duodenum [26]. During this time, fluorophore release was ~98% and ~66% from probes **13** and **16**, respectively, indicating that neither compound is sufficiently stable at pH 5.0, as shown in Figure 2B. In contrast, probes **14** and **17** showed slower fluorophore release at pH 5.0, with only 10% and 7% of free fluorophore detected at 35 min, as shown in Figure 2C.

Evaluation of the release profiles of probes **13**–**14**, **16**–**17** at pH 5.0 indicates that they followed either pseudo-first-order reaction kinetics or a genuinely first-order reaction, but additional mechanistic investigations would be needed to distinguish between these possibilities, which is beyond the scope of this initial proof of concept paper. The half-lives of probes **13**–**14** and **16**–**17** at pH 5.0 were 6.3 min, 3.5 h, 20.6 min, and 16.4 h, respectively (Table 1).

Values were recorded in triplicate and S.D was calculated.

We were unable to quantify fluorophore release from probes **13** and **16** at pH 6.5 and 7.5 using the HPLC assay, as the reaction proceeded too quickly relative to the measurement time. Instead, a fluorometric assay was performed to measure their stability under these conditions. Both probes converted to the free fluorophore within 3 min at both pH values, as shown in Appendix A and described in detail in the Appendix A. For probes **14** and **17**, their stability was measured in simulated small intestine conditions (pH 7.5) for 2 h, which is the reported transit time for this part of the GIT. Despite the fact that fluorophore release from these compounds was slower than for **13** and **16** at pH 7.5—requiring 20 and 40 min, respectively, to achieve complete fluorophore release—neither probe was stable over the full small intestine transit time. At pH 6.5, mimicking the colon, probes **14** and **17** achieved full fluorophore release after 70 and 170 min, respectively (Figure 2D). The conversion rates at both pH values followed the same pattern as above, with half-lives for probe **14** of 1.7 min (pH 7.5) and 8.1 min (pH 6.5), and for probe **17**, of 6.8 min (pH 7.5) and 28 min (pH 6.5; Table 1). In sum, the pH-sensitive linkers with two- and three-methylene spacers are not suitable for colonic delivery as the drug would be released too early and too quickly in the upper GIT. Therefore, further optimisation was necessary to delay the release of the payload until the prodrug reaches the colon.

Although the identity of the free fluorophore was confirmed by LC/MS in every case, we did not separate the cyclised side-product to confirm the hypothesised release mechanism. Nevertheless, there are many reasons to believe that the proposed intramolecular cyclisation is the relevant mechanism. The cyclisation of similar substrates is reported in the literature [21,32,33,34,35], and the rate of fluorophore release from all probes increased with pH, consistent with the assumption that the amino group is essential for the observed release. Moreover, cyclisation is reported to be faster for the five-membered ring compared with the six-membered ring due to the lower activation energy, and this also matches our results: probes **13** and **16**, with a two-methylene spacer, cyclised faster than probes **14** and **17**, containing a three-methylene spacer, at all pH values studied.

Even though the methylene spacer was introduced between the payload and the carbonate group to minimise any possible influence of the payload on the electrophilicity of the latter, we found that the stability of the pH-sensitive handles varied with different payloads. The CTG probes **13**–**14** were comparatively less stable than DCM-OH probes **16**–**17**, regardless of pH. CTG is more acidic than DCM-OH (pK_a_ 4.33 versus 7.21) [36], and as indicated in the literature for similar scaffolds, the rate of cyclisation increases with the acidity of the payload [35,37], thereby rationalising the present results. However, the observed difference in release rates was not significant in terms of colonic delivery. For example, probe **14** (with the CTG fluorophore) displayed half-lives of 1.7 min at pH 7.5 and 8.1 min at pH 6.5, while the corresponding values for probe **17** (with the DCM-OH fluorophore) were 6.8 min and 28.0 min. According to Table 1, probes **13**–**14** and **16**–**17** with CTG and DCM fluorophores completely decomposed in the small intestine simulating conditions. For this reason, we deemed that one-way ANOVA need not be processed, as these probes failed to delay the payload release in the small intestine within the reported transit time. Since these half-lives are short relative to the transit times for the small intestine and colon, respectively, the effect of the payload on the release rate was considered minimal, and CTG alone was selected for further optimisation of a representative model compound.

To optimise the release rate, probe **15** was synthesised, containing a four-methylene spacer permitting cyclisation to give a seven-membered ring. As expected, this was the most stable probe, with no detectable fluorophore release under conditions similar to those found in the stomach (pH 1.2) and duodenum (pH 5.0). Furthermore, the longer spacer protected 60% of the CTG payload from being released under small intestine-like conditions over the appropriate transit time (pH 7.5, 2 h; Figure 3). Incubating probe **15** under colon-like conditions resulted in the sustained release of >90% of the CTG payload over 38 h. The half-lives for probe **15** at pH 7.5 and 6.5 were 2.4 h and 13.2 h, respectively (Table 1). Increasing methylene spacer length still further (i.e., to five and six carbons) led to pH-sensitive linkers with very slow release profiles that were impractical for colonic delivery (98% of CTG released over 9 d and 10 d at pH 7.4, respectively). Of the three amino-AOCOM masking moieties detailed here, the one with a four-methylene spacer between the amino and carbonate groups possesses the most promising stability profile for application in prodrugs intended for colonic delivery.

### 3.4. In Vitro Release Studies of Msl Prodrugs

We chose Msl as a starting point for putting our prodrug concept into practice. Thus, Msl prodrug **21** (with a four-methylene spacer) was synthesised as its *p*-toluenesulfonic acid salt, since attempts to purify the hydrochloride salt proved futile. Several HPLC methods have been reported for the assay of Msl, using various mobile phases and a range of wavelengths [38,39,40,41,42,43,44], although none of them was suitable for the current in vitro assay due to the different pH and temperature conditions. Therefore, a new methodology was developed in which prodrug **21** was incubated in 1:1 ethanol/buffer and the HPLC analysis was run with UV detection at 310 nm, as detailed in the Appendix A.

Similarly to the probes, Msl prodrug **21** was incubated at pH 1.2 for 5 h and pH 5.0 for 35 min. The results confirmed that no Msl was released at either pH, demonstrating the good stability of the prodrug under the simulated stomach- and duodenum-like conditions. Then, the same compound was incubated at pH 7.5 and 6.5 to measure its stability under conditions mimicking the small intestine and colon, respectively. An unexpected side-product was formed that was identified by LC/MS and HRMS as **23**, the ethyl ester of Msl (Figure 4). When a standard sample of Msl was incubated under the same conditions as a control, no esterification was observed, suggesting that **23** might arise via esterification before cyclisation to release the drug. We changed the solvent to 1:1 MeCN/buffer, but, interestingly, another side-product **24** was formed with an *m/z* of 315.0 ([M+H]^+^). Identification of this side-product as **24** was supported by HRMS analysis, and a similar tendency to form this product was seen in the literature when a buffered solution of Msl was treated with Fe(II)–EDTA [45]. We believed that the formaldehyde released after cyclisation of the linker had initiated the side-reaction according to the mechanism described in detail in Appendix A and supported by literature [46] (Appendix A). Therefore decided to use 1:1 water/buffer as the solvent for subsequent assays with the Msl prodrugs.

When using water instead of ethanol or MeCN, no side products were observed, and as shown in Figure 5A, 20% of the payload was released from prodrug **21** under conditions mimicking the small intestine (pH 7.5, 37 °C) over 2 h. This release was much slower compared with probe **15**, which was considered an advantage. Moreover, the results under colon-like conditions showed 36% release of Msl after 24 h and 61% after 48 h. Further drug release was very slow and reached 93% after 6 d. We thought that decreasing the spacer length to three carbons might increase the release rate to achieve complete payload delivery within the colon transit time (30–40 h). Therefore, Msl prodrug **22** (with a three-methylene spacer) was synthesised and tested. As for **21**, prodrug **22** was fully intact after incubation at pH 1.2 for 5 h or at pH 5.0 for 35 min. It released 23.5% of Msl at pH 7.5 over 2 h (small intestine-like conditions), whereas 68% of the payload was released after 48 h at pH 6.5 (colon-like conditions), as shown in Figure 5A,B. Although complete Msl release was not achieved from either prodrug within the colon transit time, the results are still considered a step forward. First, this study provides initial proof of concept regarding the use of amino-AOCOM prodrugs for colonic delivery in an in vitro model system. It also shows that Msl prodrugs **21** and **22** successfully delayed Msl release in vitro, which should prove advantageous for delivery of the drug to the intended site of action compared with the plain Msl formulation. Thus, 77–80% of the prodrug remained intact under the simulated upper GIT conditions, and 61–68% sustained drug release was achieved under colon-like conditions, corresponding to releasing around half of the original payload at colonic pH in the model system. In contrast, unmodified Msl is absorbed rapidly in the small intestine and quickly excreted in urine (91.5% of Msl found in urine after 96 h) [47,48], confirming its limited availability to the colon. Moreover, the results obtained for the newly designed prodrugs seem comparable to those of Asacol, for which 32% of the ingested dose is absorbed and excreted in urine at 96 h [47]. There was no significant difference between the release profiles of Msl from prodrugs **21** and **22**, indicating that the linker chemistry played a minimal role compared with the model compounds. In the case of **21** and **22**, we attribute this to the carboxylic group that deprotonates within the pH range, resulting in decreasing the electrophilicity of the nearby carbonate group and subsequently decreasing its susceptibility to nucleophilic attack by the amino group. Therefore, future optimisation seemingly requires esterification of the carboxylic acid group of Msl to enhance the cyclisation step and improve the release kinetics of Msl in the colon. Based on this assumption—that the carboxylic acid group is responsible for decelerating the cyclisation of Msl prodrugs—esterification of the carboxylic group of Msl is recommended. Then, Msl release in vivo would occur in two stages, with intramolecular cyclisation of the linker to release the Msl ester followed by enzymatic hydrolysis by esterases to finally release the free drug.

## 4. Conclusions

We have designed and synthesised a series of amino-AOCOM prodrugs to improve the selectivity of current pH-dependent release systems. Initial in vitro assays identified probe **15** as stable under stomach- and duodenum-like simulated conditions, with 60% of the payload protected from release under conditions mimicking the small intestine. Furthermore, **15** released the parent drug with appropriate kinetics under colon-like conditions (>90% release over 38 h). Based on these results, Msl prodrugs **21**–**22** were also synthesised and examined for their colonic delivery potential. Release of Msl from both prodrugs was considerably slower than the model system (release of CTG from probes **14** and **15**), presumably due to the carboxylate group of Msl rendering the system zwitterionic at pH 6.5–7.5, thereby hindering the cyclisation step. These findings suggest that esterification of the carboxylic group of Msl is required to achieve faster drug release in the colon. In conclusion, this study has established the potential utility of cyclisation-activated prodrugs in targeting drugs to the colon, but further studies are needed to optimally tune the release rate of the payload from such compounds.

## Data Availability

All the data generated within this study were provided in Appendix A.

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
