# Peer review of "Design, Development, and Optimisation of Smart Linker Chemistry for Targeted Colonic Delivery—In Vitro Evaluation"

_pharmaceutics, 2023, doi:10.3390/pharmaceutics15010303_

Round 1
Reviewer 1 Report
The authors have done a interesting work presenting a future possibility to explore amino-alkoxycarbonyloxymethyl (amino-AOCOM) ether prodrugs for colonic delivery. The interesting factor of these prodrugs are that they are activated via a pH-triggered intramolecular cyclisation–elimination reaction. However, there are some minor comments which they should address before this article could be published.
1. The authors have tested these prodrug release in pH 6.5 and pH 7.5 using buffered solutions. But could the author check the same in blood serum from in-vivo or in cell culture media of pH7.5 to verify if the release outcome is the same?
2. There are some recent articles on simple linker approach to deliver drug to colon cancer using click chemistry technique. The author could discuss this to compare with their work. https://doi.org/10.1021/acsami.1c21655
3. Could the table 1 data be statistically compared using one way ANNOVA to check if the difference in time has significance.
Author Response
Dear Reviewer-1,
Many thanks for providing your comments on our manuscript entitled “Design, Development, and Optimization of Smart Linker Chemistry for Targeted Colonic Delivery—in vitro Evaluation” with submission ID: pharmaceutics-2143084. We would like to thank you for your valuable comments and suggestions to improve our manuscript overall. We have carefully gone through every comment and have provided the following response, please see the attachment. We believe that the revised manuscript and detailed responses will meet your satisfaction.
Thank you from all the authors.
Sincerely,
Corresponding authors
Prof. Jonathan Baell
Ramesh Mudududdla PhD

Reviewer 2 Report
The current investigation aimed to determine whether amino-alkoxycarbonyloxymethyl (amino-AOCOM) ether prodrugs could be a useful general strategy for the delivery of drugs to the colon in the future. These prodrugs are activated by a pH-triggered intramolecular cyclisation-elimination reaction rather than enzymes, which eliminates the potential for inter- and intra-subject variations in enzyme activities. Model compounds were synthesized and tested in acidic and alkaline environments to demonstrate the concept's viability (GIT). Excellent stability was shown by Probe 15 in conditions mimicking the stomach and duodenum, and 60% of the payload was safe in conditions mimicking the small intestine. Furthermore, 15 of them showed sustained release at colonic pH, releasing >90% of their payload over a period of 38 hours. In a simulated upper GIT environment, prodrugs 21 and 22 of mesalamine (Msl) were more stable than probe 15, but their release was slower (61% to 68%) at colonic pH over 48 hours. Both of these prodrugs had a release profile that was on par with Asacol, a commercially available product. This study provides proof of concept for the use of a cyclization-activated prodrug for colon delivery and also suggests that release characteristics still vary on an individual basis.
1- The objective of the present study was to explore the potential of amino-alkoxycarbonyloxymethyl (amino-AOCOM) ether prodrugs as a general approach for future colonic delivery. This approach is used first time? There are some studies related to this approach and its better if authors can explain the novelty of their work by comparing with previous studies in introduction section as research question is not very clear for reader.
2-Why NMR data is not provided in detail?
3-Test solutions of probes 13–17 were prepared at 25 µM in a 1:1 mixture of ethanol and the 104 respective buffer. How this concenteration was selected? It should be mentioned.
4-Results section has no details on the results of synthesis characterization procedures?
5- There are lot of typo errors and extensive deep reading is required.
Author Response
Dear Reviewer-2,
Many thanks for providing your comments on our manuscript entitled “Design, Development, and Optimization of Smart Linker Chemistry for Targeted Colonic Delivery—in vitro Evaluation” with submission ID: pharmaceutics-2143084. We would like to thank you for your valuable comments and suggestions to improve our manuscript overall. We have carefully gone through every comment and have provided the following response, please see the attachment. We believe that the revised manuscript and detailed responses will meet your satisfaction.
Thank you from all the authors.
Sincerely,
Corresponding authors
Prof. Jonathan Baell
Ramesh Mudududdla PhD

Round 2
Reviewer 2 Report
I appreciate the positive reactions to my suggestions.